# Food Waste Generation in Germany in the Scope of European Legal Requirements for Monitoring and Reporting

**Dominik Leverenz [1,\*], Felicitas Schneider [2] , Thomas Schmidt [2], Gerold Hafner [1], Zuemmy Nevárez [1] and Martin Kranert [1]**

1   Institute for Sanitary Engineering, Water Quality and Solid Waste Management, University of Stuttgart, Bandtäle 2, 70569 Stuttgart, Germany; gerold.hafner@iswa.uni-stuttgart.de (G.H.); st169724@stud.uni-stuttgart.de (Z.N.); martin.kranert@iswa.uni-stuttgart.de (M.K.)
2   Thünen—Institute of Market Analysis, Federal Research Institute for Rural Areas, Forestry and Fisheries, Bundesallee 63, 38116 Braunschweig, Germany; felicitas.schneider@thuenen.de (F.S.); thom-as.schmidt@thuenen.de (T.S.)
\*   Correspondence: dominik.leverenz@iswa.uni-stuttgart.de; Tel.: +49-711-685-658-16

**Abstract:** The European Commission and the German government are committed to the United Nations SDG target 12.3 of reducing food waste along production chains and halving it at retail and consumer levels by 2030. European member states are required to monitor national food waste levels and report annual progress to the European Commission from 2020 onward. In this regard, the main objective of our study is to provide food waste quantities for Germany by applying methods meeting the legal requirements for monitoring and reporting within Europe-wide harmonized methodology. Our results refer to 2015 and are based on the best available data, using a combination of official statistics, surveys, and literature. We found that approx. $11.9 \pm 2.4$ million tonnes $(144 \pm 28 \text{ kg/(cap·year)})$ of food waste were generated in Germany, while the reduction potentials varied throughout the different sectors. Even though the underlying data show uncertainties, the outcome of the study represents a starting point for the upcoming monitoring activities in Germany by uncovering data and knowledge gaps. To meet the political reduction targets, a national food waste strategy was launched in 2019 by the German Federal Ministry of Food and Agriculture, which is an important step toward cooperation and exchange between actors along the entire food chain, raising awareness, and improving data quality, monitoring, and implementation of prevention measures in practice.

**Keywords:** food waste; SDG 12.3; monitoring; reporting; legal framework; Delegated Decision (EU) 2019/1597; baseline 2015; common methodology; in-depth measurements; National Strategy for Food Waste Reduction

## 1. Introduction

The 2030 Agenda for Sustainable Development of the United Nations is a commitment to achieving sustainable development worldwide by 2030. This global action plan formulates 17 sustainable development goals (SDGs) and 169 specific targets based on the three dimensions of sustainable development: economy, society, and environment. SDG 12 is dedicated to ensuring sustainable production and consumption patterns. The specific target 12.3 covers food loss and waste reduction and calls for halving global food waste at the retail and consumer levels, as well as reducing food losses along production and supply chains, including post-harvest losses [1].

### 1.1. Food Waste Quantities in Different Countries

In 2011, the FAO commissioned a study that examined global food waste levels and reported that around one-third of the global food production ($\approx$1.3 billion t/year) is either lost or wasted [2]. The authors of the study emphasized that industrialized nations

waste significantly more food than developing countries, particularly at the consumer level. For instance, consumers in Europe and North America generate between 95 and 115 kg/(cap·year) of food waste, whereas in sub-Saharan Africa and Southeast Asia, only between 6 and 11 kg/(cap·year) is generated. Gustavsson et al. (2011) had to make several assumptions due to major research gaps, insufficient input data, and statistical uncertainties. Hence, Gustavsson et al. stated that the aforementioned estimations on food waste quantities must be interpreted with considerable caution [3]. Stenmarck et al. estimated food waste quantities in Europe at approximately 88 million t/year with a moderately high uncertainty of ±14 million tonnes, or ±16% [4]. This uncertainty is a result of a relatively small number of studies providing sufficient quality of data and varies regarding the different sectors: primary production, processing, wholesale and retail, restaurants and food services, and households. In June 2020, the European Commission published guidance on reporting of data on food waste and food waste prevention according to Commission Implementing Decision (EU) 2019/2000 [5]. In addition to general requirements for the monitoring and reporting, the guidance provides good-practices examples for the collection of food waste data from different European countries such as Finland, Sweden, Norway, Denmark [6], Germany [7], the United Kingdom [8], Sweden [9], Switzerland [10], and Austria [11]. In 2012, however, Kranert et al. conducted the first study on food waste in Germany and estimated that approximately 11 million tonnes of food waste were generated in food processing, retail and other distribution of food, restaurants and food services, and households (excluding primary production). In addition to the estimation of food waste quantities, approaches and recommendations to reduce food waste were elaborated. Furthermore, the study emphasized the need for future research to fill data gaps and improve data quality [12]. Since 2012, food waste has been an issue of growing public concern, which has resulted in steadily increasing scientific interest. To generate knowledge on this matter, international literature has focused on data collection across different sectors of the food supply chain, improving data accessibility and quality [13]. In Germany, several studies contributed to the knowledge gain by collecting food waste data and investigating reduction strategies, particularly at the consumer level [7,14–19].

### 1.2. Food Waste Reduction Initiatives

Several studies and literature reviews have identified a lack of information and findings concerning the implementation and evaluation of food waste prevention measures [20–23]. Priefer et al., for example, addressed the need to investigate the impact of economic and regulatory instruments, because most reduction measures implemented in the past have been soft instruments such as awareness campaigns, round tables, networks, and information platforms. Stöckli et al. stated that informational interventions are the most popular type of intervention for reducing food waste, although they often do not lead to the desired result. According to a recent review paper, the literature promotes and recommends initiatives such as cooking classes, fridge cameras, food sharing apps, advertising, and information sharing, but provides little or no robust evidence on their effectiveness [23]. Reynolds et al. described this situation as worrying, as these recommendations are being proposed as successful approaches. However, except for a few studies, there is no reproducible evidence quantified to ensure credibility. Goossens et al. stated that many of the proposed reduction measures are incomplete with regard to their economic, environmental, or social assessments, and that efficiency is only rarely calculated. This causes a certain complexity for practitioners and decision-makers when distinguishing measures according to their efficiency and prioritizing them for future implementations [24]. With reference to these findings, it becomes evident that research needs to generate reliable information to implement monitoring of food waste quantities and reduction measures.

On a political level, food waste reduction initiatives from the UK can serve as a role model. The UK succeeded in reducing the amount of household food waste by approx. 1.44 million tonnes in 2018 compared with 2007. Accordingly, food waste from private households decreased from 132 kg/(cap·year) to approx. 100 kg/(cap·year) [25]. The

starting point for this positive development was set in 2000, when the Waste and Resources Action Program (WRAP) was established to support sustainable waste management and increase recycling in the UK. To follow up on their recycling initiatives and to promote understanding of food waste issues to the general public, WRAP launched a campaign called Love Food Hate Waste in 2007. The objective was to encourage consumers to reduce food waste through awareness-raising information [26]. The campaign was one of the first of its kind worldwide and raised awareness not only among the general public, but also among stakeholders from industry, politics, and science. In the following years, numerous initiatives to reduce food waste in all parts of the food supply chain were implemented throughout the UK and several other countries. However, no other country can present a similarly positive trend in reducing food waste to date. Nevertheless, WRAP claims that even more measures are needed to ensure that most people and organizations become involved and implement the necessary changes to achieve political targets.

### 1.3. Food Waste Reduction Targets in Europe and Definitional Framework

In line with SDG 12.3, the European Union (EU) proposed a target of food waste reduction as well as a food waste definition in its revised waste framework directive [27].

In 2020, the European Commission adopted a new Circular Economy Action Plan, which considers the reduction in food waste as a key action under the current EU Farm-to-Fork Strategy [28]. Figure 1 illustrates differences and similarities in the definitional framework between the UN and the EU with regard to SDG 12.3.

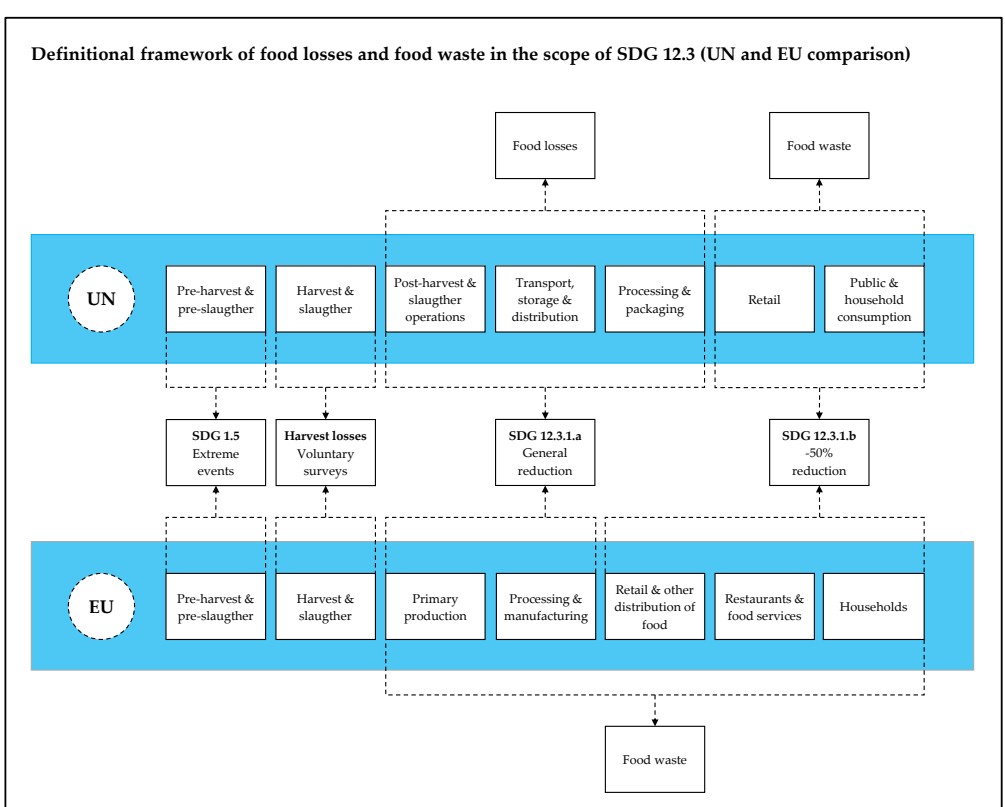

**Figure 1.** Definitional framework of food losses and food waste in the scope of SDG 12.3. Comparison between the UN [29,30] and EU [31].

Unlike the United Nations, European legislation consistently defines discarded food as "food waste" throughout the food supply chain instead of applying the term "food losses" in earlier stages. In addition to terminology, methodological differences in data collection exist as well. The Food and Agriculture Organization of the United Nations (FAO), for example, uses general food balance sheets, whereas the European member states have

committed to conducting additional in-depth measurements every fourth year starting in 2020. In both approaches, the entire food supply chain will be monitored according to Figure 1. Harvest losses, however, are treated separately and can be reported by UN member states on a voluntary basis [30,32].

### 1.4. Political Framework in Europe

The European Waste Framework Directive 2008/98/EC was amended by Directive (EU) 2018/851 of the European Parliament and Council on 30 May 2018. An EU-wide target has been adopted to meet SDG 12.3 by 2030 and to achieve a midterm goal of a 30% food waste reduction by 2025. Further amendments are related to requirements for measuring, monitoring, and reporting food waste within the EU as follows:

- Article 3 (4a) defines food waste as all food that has become waste as defined in European food law [33];
- Article 9 (1g) explicitly commits to SDG 12.3 and calls for the implementation of food waste reduction measures along the entire food value chain;
- Article 9 (1h) encourages food donations and other forms of redistribution of food primarily for human consumption and next for animal feed or reprocessing into non-food products. Annex IVa (3) further recommends providing fiscal incentives for food donations as a possible economic instrument;
- According to Article 9 (5), member states shall monitor food waste based on a common methodology established in Delegated Decision (EU) 2019/1597 [31];
- By 31 March 2019, the commission shall adopt the Delegated Decision that establishes a common methodology and minimum quality requirements for the uniform measurement and reporting of food waste levels (Article 9 (8) and Article 37 (7));
- Article 29 (2a) requires member states to adopt specific food waste prevention programs within their obligatory waste prevention programs;
- Article 37 (3) obliges member states to report annually on their food waste quantities and trends, starting with the reference year 2020.

To achieve the food waste reduction targets and ensure a high level of stakeholder contribution, the European Commission initiated an interdisciplinary platform on food losses and food waste. The platform established in 2016 aims to support all actors during measuring, reducing, monitoring, and reporting food waste. Members include key actors that represent both public and private interests such as international organizations, EU institutions, member states, and private sector organizations [34]. One result of this cooperation was the formulation of recommended actions to prevent food waste [35].

### 1.5. Timeline and Methodology for the EU Food Waste Reporting

Based on the EU legal framework, a preliminary timeline for reporting food waste in Europe can be expected as shown in Figure 2. Member states shall prepare annual reports of the generated food waste and report them to the European Commission. In-depth measurements are required in minimum four-year intervals for each sector of the value chain. The data may be updated in the interim using latest available waste coefficients as well as indicators that are correlated with official statistics such as food production in agriculture (primary production), fishery and hunting, production of processed food, turnover of food products, employment, population, and disposable household income [31]. The legal framework was adopted in May 2019 by the Commission Delegated Decision (EU) 2019/1597, supplementing Directive 2008/98/EC of the European Parliament and Council regarding a common methodology and minimum quality requirements for the uniform measurement of food waste levels. Moreover, the Implementing Decision (EU) 2019/2000 regulates the format for reporting data on food waste and submitting quality assessments [36].

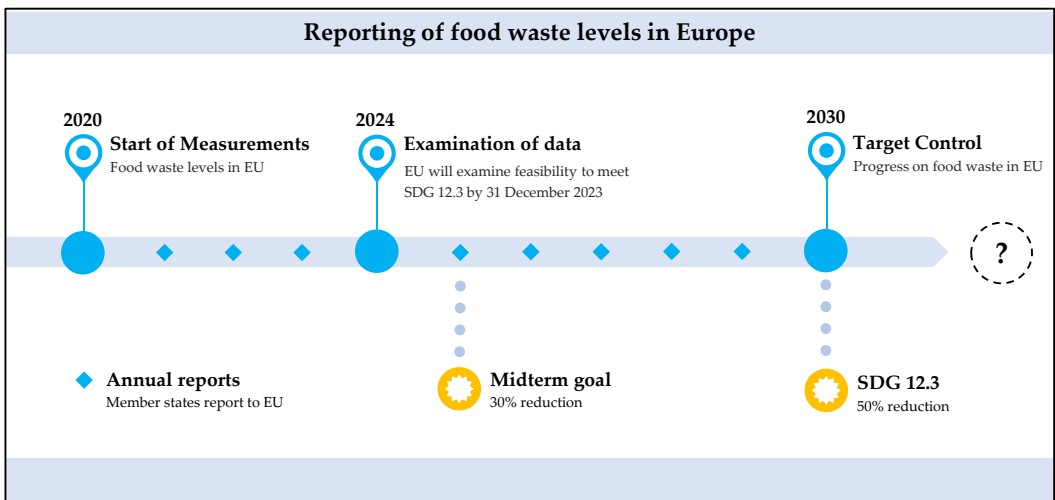

**Figure 2.** Preliminary timeline for reporting food waste levels in Europe until 2030 based on requirements from Directive (EU) 2018/851.

The Delegated Decision (EU) 2019/1597, however, determines that food waste will be reported separately for five sectors of the value chain according to their NACE codes (the industry standard classification system used in the EU): primary production, processing and manufacturing, retail and other distribution of food, restaurants and food services, and households (Article 1). In addition to food waste terminology, there is a distinction of what is not considered food waste. Food waste is defined as all food that has become waste, including inedible parts. Additionally, edible plants that have not been harvested or by-products from the production of food are not considered food waste. However, member states shall measure the annual amount of food waste in tonnes of fresh mass. Article 4 of this Delegated Decision provides the minimum quality requirements to ensure the reliability and accuracy of data. Measurements should be based on a representative sample of the population and adequately reflect variations. In terms of representativeness, the Delegated Decision vaguely defines the meaning and size of a representative sample. Annex IV (a and b) further loosens these requirements by stating that calculations should refer to the best available data when in-depth measurements are not used.

### 1.6. Objectives

With the present calculation of food waste quantities in Germany, we aimed to provide a starting point for the future reporting and monitoring of food waste toward 2030. The main objective of the study was to present food waste quantities for each sector of the supply chain according to Delegated Decision (EU) 2019/1597, which requires member states to conduct in-depth measurements every four years. To assess the quality of the best available data for the reporting and monitoring process in Germany, we report statistical uncertainties and identify data gaps that need to be filled in the future.

With reference to the Special Issue of *Sustainability* (ISSN 2071-1050), "National Food Loss and Waste Prevention Strategies and Monitoring Approaches—an Interdisciplinary Challenge for Decision Makers, Researchers and Practice", we aimed to address and discuss following research questions:

(1) What are the food waste reduction potentials in Germany and how do the results of our study relate to SDG 12.3?
(2) Should the focus be on preventing and halving all food waste or should the priority be on edible and avoidable parts?
(3) Should all food and drinks, including all disposal paths and alternative use for animal feed, industrial use, and non-food purposes, be considered in food waste monitoring?
(4) How should the food waste calculations at the national level be interconnected with the reduction strategy and monitor significant developments?

## 2. Materials and Methods

### 2.1. Scope of the Study and Definitional Framework

For the remainder of this paper, we use the term food waste according to the legal framework that defines food waste as all food that has become waste within the Directive (EU) 2018/851 of the European Parliament and of the Council of 30 May 2018 [27].

We investigated food waste quantities across the entire German food supply chain. Figure 3 provides a graphical overview of the common methodology for in-depth measurements of food waste in Europe that comply with Annex III of the Commission Delegated Decision (EU) 2019/1597. Accordingly, we used the best available data sources, while they are consistent with the options listed in Annex III of the Commission Delegated Decision (cf. Figure 3). Due to a lack of data on food waste produced by German retail and other distributers of food, methodological compromises were necessary. Hence, food waste coefficients proved to be the best available data and were used to calculate the estimates for the retail sector. However, the study's findings provide a first contribution to the recently formulated legal requirements by presenting food waste quantities and reduction potentials in Germany across the entire food supply chain.

| Measurement methods | Primary production | Processing and manufacturing | Retail and other distribution of food | Restaurants and food services | Households |
|---|---|---|---|---|---|
| Direct measurement | | | | ✓ | |
| Mass balance | ✓ | | | ▮ | ▮ |
| Waste compositional analysis | | | | ✓ | ✓ |
| Questionnaires and interviews | | ✓ | ▮ | ▮ | ▮ |
| Coefficients and production statistics | ✓ | ✓ | ✓ (▮) | ▮ | ▮ |
| Counting or scanning | | | | | ▮ |
| Diaries | ▮ | ▮ | ▮ | | ✓ |

| | |
|---|---|
| ☐ | Measurement methods that are foreseen to monitor food waste levels in Europe |
| ✓ | Measurement methods used to investigate food waste levels in Germany |
| ▮ | Measurement methods that are not foreseen to monitor food waste levels in Europe |

**Figure 3.** Common methodology for in-depth measurement of food waste in Europe according to Annex III of the Commission Delegated Decision (EU) 2019/1597.

### 2.2. Methodical Approach and Data Basis

The preparatory work for the present study is based on national research projects that we have conducted since 2012 [12,37–39]. The data basis and the methodological approach gradually enhanced over time and were adapted to the requirements of the Delegated Decision within a technical report in 2019 [7]. The data basis of our study for each sector of the food supply chain is presented in the Supplementary Materials. We used different data sources such as official harvesting and production statistics (Tables S1 and S2), food waste coefficients from the literature (Tables S1, S4 and S6), food waste coefficients generated via surveys and questionnaires (Tables S2–S4), nutritional databases (Table S1), structural data for different restaurants and food services from statistics and literature (Table S5), information from waste compositional analyses (Table S7), household diaries (Table S8), and official waste statistics [40]. The study was calculated for the year 2015 and all available data sources originated from that year or the most proximate year. In addition to the obligatory reporting of total food waste quantities in fresh mass, we calculated data sets that may be reported on voluntary basis according to Delegated Decision 2019/1597 as far

as information was available. For instance, this includes the avoidable percentages of food waste along the different sectors of the food supply chain and food waste from households disposed down the drain.

*2.3. Production and Supply Chains*

2.3.1. Primary Production

The legal definition of food waste from the revised Waste Framework Directive 2009 as amended by Directive 2018/851 refers to waste from post-harvest agricultural processes. Losses during the harvest of agricultural commodities and the rearing of animals are not considered food waste according to the legal definition in Europe. Consequently, losses of agricultural commodities during harvest processes are not included in the official harvesting statistics. In our study, we therefore focused on the quantification of food waste in agricultural production during post-harvest processes.

The produced food quantities (harvested quantities ($h_{PP,i}$)) in primary production are provided within the national harvesting statistics (cf. Table S1). Official waste statistics from primary production, however, do not provide waste codes that identify food waste as defined by the revised Waste Framework Directive [27]. Therefore, we estimated the food waste quantities in German primary production using waste coefficients from the literature ($wc_{PP,i}$), which indicate the percentage of waste in relation to the harvested food quantities (cf. Table S1). The quantity of food waste during post-harvest processes such as handling, storage, and transportation is determined using Equation (1). According to the available data sources, the confidence interval for the average food waste quantities is calculated within the limits of the absolute min. and max. values (mean $\pm$ CI$_{\mathrm{Min,Max}}$):

$$FW_{PP} = \sum_i h_{PP,i} \cdot wc_{PP,i} \tag{1}$$

where $FW_{PP}$ is the quantity of food waste during post-harvest processes, storage, and transport in primary production ($PP$) in t/year; $h_{PP,i}$ is the harvested quantities for selected food categories ($i$) in German primary production in t/year (cf. Table S1); and $wc_{PP,i}$ is the waste coefficients for selected food categories ($i$) in German primary production in mass % (cf. Table S1).

Similar to Gustavsson et al., we calculated agricultural food waste separately for the food categories of cereals, potatoes (roots and tubers), sugar beets, oilseeds, pulses, fruits, vegetables, meat, fish, milk, and eggs. To estimate the avoidable quantities, we subtracted the average percentage of non-edible components (e.g., peels and bones) from the food waste quantities for each food group. The average percentages of peels and bones for different food categories are presented in Table S1.

2.3.2. Processing and Manufacturing

The produced food quantities ($p_{PM,i}$) by the German processing and manufacturing sector are provided within the national production statistics and classified according to official NACE codes. We estimated the food waste quantities for each subsector of the food processing and manufacturing sector using waste coefficients ($wc_{PM,i}$), which indicate the percentage of waste in relation to the produced food quantities in Germany (cf. Table S2). The average food waste quantities are calculated within the 95% confidence interval (mean $\pm$ CI$_{\alpha=0.05}$) using Equation (2):

$$FW_{PM} = \sum_i p_{PM,i} \cdot wc_{PM,i} \tag{2}$$

where $FW_{PM}$ is the quantity of food waste in the German food processing and manufacturing sector ($PM$) in t/year; $p_{PM,i}$ is the production quantity for different segments ($i$) of the German food processing and manufacturing sector in t/year (cf. Table S2); $wc_{PM,i}$ is the waste coefficients for different segments ($i$) of the German food processing and manufacturing sector in mass % (cf. Table S2).

In the absence of essential data sources, a questionnaire was sent to the members of the Federation of German Food and Drink Industries and the Chamber of Industry and Commerce. A total of 3475 companies were surveyed, of which 118 responded to the questionnaire (response rate: 3.4%). For the present study, 100 questionnaires provided useful data from which food waste coefficients could be derived. With regard to the basic population, i.e., the German processing and manufacturing sector, the sample size does not meet the requirements of representativeness, but it nevertheless represents the best available data at present. The survey also investigated drivers that cause food waste. Based on the identified drivers (cf. Figure S1), we assumed that process-inherent losses (e.g., retained samples stored for quality assurance, technical defects, or shrinkage) lead to food waste that cannot be controlled or reduced. In contrast, we assumed that food waste related to wrong process management, damage and spoilage, overproduction, returns from retailers, and faulty batches can be controlled and reduced. Consequently, we assumed that approx. 55% of food waste in the food processing and manufacturing sector can theoretically be avoided.

### 2.4. Retail and Consumer Levels

#### 2.4.1. Retail and Other Distribution of Food

Food waste in retail and other distribution of food was evaluated on the basis of food waste coefficients that were published in the sustainability report of a German supermarket chain [41]. In addition, waste balance data from 77 stores of a second supermarket chain were provided directly to the authors (cf. Table S3). Similar to the processing and manufacturing sector, the sample size ($n = 77$) of the retail sector does not meet the requirements of representativeness, but represents the best available data at present. The coefficients describe the average amount of food waste per square meter of sales area. Equation (3) describes the calculation for the estimation of food waste in the food retail sector based on specific waste coefficients ($wc_R$) and the sales area ($a_R$). The average food waste quantities in retail were calculated within the 95% confidence interval (mean $\pm$ CI$_{\alpha = 0.05}$):

$$FW_R = \frac{a_R \cdot wc_R}{1000} \tag{3}$$

where $FW_R$ is the quantity of food waste in the German food retail sector ($R$) in t/year; $a_R$ is the food retail sales area: 35.6 million m$^2$ in 2015 [42]; $wc_R$ is the waste coefficients for the German retail sector per square meter of sales area and year (cf. Table S4).

The amount of food waste in wholesale markets is based on surveys conducted by the German Wholesale Markets Association [43]. For this purpose, the annual turnover of goods is multiplied with the percentage of the mean organic waste quantities generated (cf. [7]). Lebersorger and Schneider showed in an Austrian case study (n = 612) that about 28% of all food waste from the food retail sector occurs because of reaching the best before date. Another 56% of the disposed food was sorted out due to apparent defects [44]. Assuming transferability to the German food retail sector, about 84% of food waste from retail can potentially be avoided.

#### 2.4.2. Restaurants and Food Services

Biodegradable waste from restaurants and food services is listed in the official waste balance sheets of the Federal Statistical Office under waste code 200,108 [40]. In addition to the reported waste statistics, there are beverages and liquid food waste such as soups and sauces that are disposed through the sewer system, as well as solid food waste that is not collected as part of the separate collection of kitchen and canteen waste and is instead disposed, for example, through the municipal residual waste collection system and is thus included in the latter statistics. We used literature values to estimate the share of waste from restaurants and food services based on waste compositional analyses.

To obtain a more accurate approximation of the true food waste quantities, calculations for different restaurants and food services were conducted on the basis of specific waste

coefficients to extend the official waste balance sheets. This procedure is in line with Kranert et al. and was separately applied for full-service restaurants, event gastronomy, fast-food restaurants, the accommodation sector, hospitals (care sector), schools, colleges, childcare facilities, business canteens, the German armed forces, and prisons. Food waste for restaurants and food services was calculated by combining structural data ($s_{RFS,i}$) and waste coefficients ($wc_{RFS,i}$) for each facility type (Equation (4)). A summary of the food waste coefficients used from the literature and their avoidable components is provided in Table S6. Accordingly, the confidence interval for the food waste quantities was calculated within the limits of the absolute min. and max. values (mean $\pm$ $CI_{Min,Max}$):

$$FW_{RFS} = \sum_i s_{RFS,i} \cdot wc_{RFS,i} \tag{4}$$

where $FW_{RFS}$ is the quantity of food waste in German restaurants and food services (*RFS*) in t/year; $s_{RFS,i}$ is the structural data for different restaurants and food services (*i*) (cf. Table S5); $wc_{RFS,i}$ is the waste coefficients for different restaurants and food services (*i*) (cf. Table S6).

2.4.3. Households

To determine food waste in households, the relevant disposal paths of the municipal waste collection system (residual and biowaste) as well as the other disposal paths (home composting, sewers, and animal feed) were investigated. The quantity of residual and biowaste from separate collection in households ($s_{HH,i}$) is published by the Federal Statistical Office on an annual basis [45]. The percentage of food waste in the municipal waste collection system (residual waste and bio waste bin ($wc_{HH,i}$)) was determined on the basis of waste compositional analyses. To estimate food waste in other disposal paths, information from a diary-based study with a representative panel (~7000 German households) was used. Participants had to estimate the type and quantity of food waste and decide whether the waste was avoidable or unavoidable based on their individual and subjective perceptions. With reference to the study's finding, we assumed that approximately 44% of household food waste is avoidable [15]. To verify our findings in households, we calculated two scenarios. In Scenario I, we used a combined method to estimate the food waste quantities in households. Waste statistics and information from waste compositional analyses were used to calculate the quantity in municipal collection, while the information from household diaries provided insights into the relevance of the additional disposal paths. Thus, food waste in German households was calculated using Equation (5) for Scenario I. According to the available data sources (Tables S7 and S8), the confidence interval for the average food waste quantities in households was calculated within the limits of the absolute min. and max. values (mean $\pm$ $CI_{Min,Max}$). In Scenario II, we used the estimates of the diary-based panel study of Hübsch and Adlwarth and compared the results with Scenario I:

$$FW_{HH} = \frac{\sum_i s_{HH,i} \cdot wc_{HH,i}}{r_{FW}} \tag{5}$$

where $FW_{HH}$ is the quantity of food waste in German households in kg/(cap·year); $s_{HH,i}$ is the quantity of household residual and bio waste from separate collection in kg/(cap·year) (cf. Table S7); $wc_{HH,i}$ is the waste coefficients describing the share of food waste in the residual and biowaste bins of households in mass % (cf. Table S7); $r_{FW}$ is the relative relation between the quantity of food waste in the municipal waste collection system (residual and biowaste bins) and the total amount of food waste from households (including other disposal paths) in mass % (cf. Table S8).

## 3. Results

### 3.1. Food Waste in Production and Supply Chains

Table 1 presents the food waste quantities in German primary production and supply chains, including post-harvest processes such as storage, transport, processing, and manufacturing.

**Table 1.** Food waste quantities in German primary production and supply chains (processing and manufacturing) in 2015 (1000 metric tonnes of fresh mass).

| Sector | Food Groups | Food Waste | Avoidable Parts |
|---|---|---|---|
| | Cereals | $202 \pm 64$ | $202 \pm 64$ |
| | Potatoes | $243 \pm 85$ | $174 \pm 25$ |
| | Sugar (from sugar beets) | $140 \pm 84$ | $140 \pm 84$ |
| | Oilseeds | $2 \pm 1$ | $1 \pm 0.3$ |
| | Pulses | $0.1 \pm 0.04$ | $0.07 \pm 0.02$ |
| Primary production | Fruit | $113 \pm 33$ | $100 \pm 29$ |
| | Vegetables | $177 \pm 31$ | $133 \pm 53$ |
| | Meat | $185 \pm 85$ | $150 \pm 25$ |
| | Fish | $61 \pm 28$ | $32 \pm 6$ |
| | Milk (dairy products) | $196 \pm 67$ | $196 \pm 67$ |
| | Eggs | $43 \pm 7$ | $37 \pm 9$ |
| $\sum$ Food waste in primary production | | $1360 \pm 485$ | $1165 \pm 362$ |
| Manufactured products (NACE Code) | | | |
| | Meat and meat products (10.1) | $32 \pm 29$ | $18 \pm 16$ |
| | Fish, crustaceans and mollusks (10.2) | $16 \pm 7$ | $9 \pm 4$ |
| | Fruit and vegetables (10.3) | $128 \pm 59$ | $70 \pm 32$ |
| | Vegetable and animal oils and fats (10.4) | $5 \pm 4$ | $3 \pm 2$ |
| Supply chains (processing and manufacturing) | Dairy products (10.5) | $219 \pm 114$ | $120 \pm 63$ |
| | Grain mill products and starch products (10.6) | $9 \pm 7$ | $5 \pm 4$ |
| | Bakery and farinaceous products (10.7) | $687 \pm 135$ | $378 \pm 74$ |
| | Other food products (10.8) | $205 \pm 80$ | $113 \pm 44$ |
| | Beverages (11.0) | $865 \pm 315$ | $476 \pm 173$ |
| $\sum$ Food waste in processing and manufacturing | | $2166 \pm 750$ | $1191 \pm 413$ |
| $\sum$ Food waste in production and supply chains | | $3526 \pm 1235$ | $2356 \pm 775$ |
| SDG 12.3 and EU target | | General reduction | |

In German primary production, approximately 1.4 ($\pm 0.5$) million tonnes of food waste were generated in 2015, of which about 86% (1.2 ($\pm 0.4$) million tonnes) could have been used for human consumption and thus theoretically been avoided. The reduction potential was estimated by subtracting the share of peels and bones from the food waste quantities (approx. 195,000 tonnes of peels and bones). The estimates for the food processing and manufacturing industry were based on a company survey (n = 100) and resulted in an average of 2.2 ($\pm 0.8$) million tonnes of food waste in 2015, of which about 55% (1.2 ($\pm 0.4$) million tonnes) represents an avoidable potential. The overall amount of food waste in German production and supply chains was approximately 3.6 ($\pm 1.3$) million tonnes in 2015. The avoidable parts amounted to 2.4 ($\pm 0.8$) million tonnes.

### 3.2. Food Waste at Retail and Consumer Levels

An overview of the food waste quantities at the German retail (including other distribution of food) and consumer levels in 2015 is provided in Table 2.

**Table 2.** Food waste quantities at the German retail (including other distribution of food) and consumer levels in 2015 (1000 metric tonnes of fresh mass).

| Sector | Market | | Food Waste | Avoidable Parts |
|---|---|---|---|---|
| Retail and other distribution of food | Retail | | 434 ± 134 | 365 ± 113 |
| | Wholesale market | | 65 ± 22 | 54 ± 18 |
| ∑ Food waste in retail and wholesale market (excl. food donations) | | | 499 ± 155 | 419 ± 130 |
| SDG 12.3 and EU target (−50% by 2030) | | | ≈250 (−50%) | ≈210 (−50%) |
| Facilities | | | | |
| Restaurants and food services | Full-service restaurants | | 414 ± 67 | 213 ± 43 |
| | Event gastronomy | | 282 ± 46 | 145 ± 29 |
| | Fast-food restaurants | | 145 ± 26 | 104 ± 21 |
| | Accommodation sector | | 80 ± 21 | 58 ± 11 |
| | Hospitals | | 64 ± 11 | 52 ± 11 |
| | Schools | | 49 ± 6 | 38 ± 9 |
| | Childcare facilities | | 68 ± 8 | 53 ± 12 |
| | Colleges and universities | | 74 ± 12 | 51 ± 10 |
| | Care institutions | | 134 ± 22 | 108 ± 22 |
| | Business canteens | | 298 ± 54 | 242 ± 66 |
| | German Armed Forces | | 8 ± 1 | 4 ± 1 |
| | Prisons | | 18 ± 3 | 9 ± 2 |
| ∑ Food waste in restaurants and food services | | | 1634 ± 277 | 1077 ± 237 |
| SDG 12.3 and EU target (−50% by 2030) | | | ≈817 (−50%) | ≈539 (−50%) |
| Disposal paths | | | | |
| Households | Municipal waste collection | Residual waste bin | 3336 ± 304 | 1468 ± 134 |
| | | Biowaste bin | 1742 ± 156 | 767 ± 69 |
| | Other disposal paths | Home composting | 633 ± 58 | 278 ± 25 |
| | | Pet feeding | 353 ± 99 | 155 ± 43 |
| | | Sewers | 789 ± 271 | 347 ± 119 |
| | | Others | 115 ± 17 | 51 ± 51 |
| ∑ Food waste in households (excluding sewers) | | | 6179 ± 634 | 2719 ± 322 |
| SDG 12.3 and EU target (−50% by 2030) | | | ≈3090 (−50%) | ≈1360 (−50%) |
| ∑ Food waste at the German retail (including other distribution of food) and consumer levels in 2015 | | | 8312 ± 1066 | 4215 ± 689 |
| SDG 12.3 and EU target (−50% by 2030) | | | ≈4156 (−50%) | ≈2108 (−50%) |

The food waste estimates in retail are based on physical data from a survey of food retailers (*n* = 77). Accordingly, the average food waste quantity in the German retail sector was approx. 434,000 (±134,000) tonnes in 2015 (Table 2). The amount of food waste generated by German wholesale markets was estimated as 65,000 (±22,000) tonnes in 2015. Consequently, food waste from both sectors totaled an average of 499,000 (±155,000) tonnes in 2015, excluding food donations. In addition, 200,000 tonnes of food were donated to food banks and not considered food waste [46]. The theoretical reduction potential in the German retail sector amounts to 419,000 (±130,000) tonnes for 2015 (Table 2). However, ambiguities exist regarding the interpretation of food waste from wholesale markets between the UN and the EU. According to UNEP's Food Waste Index, waste streams from wholesale markets are considered food losses [29] and should be therefore part of Section 3.1. In contrast, the EU measures wholesale waste streams as food waste [31].

In the German restaurants and food services, approx. 1.6 (±0.3) million tonnes of food waste were generated in 2015. On average, approx. 1.1 (±0.2) million tonnes were considered to be avoidable food waste. The food waste estimates from restaurants and food services can be verified by official statistics. According to data in waste statistics, approx. 928,000 tonnes of biodegradable waste from kitchens and canteens were collected separately from municipal household waste and transported to waste treatment plants [40]. In addition, approx. 829,000 (±76,000) tonnes of food waste from restaurants and food

services were disposed of together with the municipal household waste collection [7]. Resultantly, a total of 1.8 (±0.1) million tonnes of food waste from restaurants and food services ended up in the municipal waste collection system in 2015, a quantity that lies within the range of the estimated quantities in Table 2. In households, approx. 5.1 (±0.5) million tonnes of food waste were disposed through the municipal waste collection system: residual household waste bin (3.3 ± 0.3 million tonnes) and biowaste bin (1.7 ± 0.2 million tonnes). Another 1.9 (±0.5) million tonnes of food waste ended up in other disposal paths. Other disposal paths refer to home composting (633,000 ± 58,000 tonnes), feeding food waste to pets (353,000 ± 99,000 tonnes), disposing of mainly liquid food waste through the sewer system (789,000 ± 271,000 tonnes), and others such as central collection points for recyclables (115,000 ± 17,000 tonnes). According to the Delegated Decision (EU) 2019/1597, there is no reporting obligation for food waste, which is disposed through the sewer system. Thus, approx. 6.2 (±0.6) million tonnes of food waste from German households (excluding sewers) were generated in 2015, of which about 44% could have been avoided. In contrast, private feeding of food waste to pets does not meet the exception of the Delegated Decision and was therefore included in the values.

The overall amount of food waste at the German retail (including other distribution of food) and consumer levels was approximately 8.3 (±1.1) million tonnes in 2015. The theoretically avoidable parts amounted to 4.2 (± 0.7) million tonnes. To achieve SDG 12.3 and EU reduction targets (−50%), more than about 4.2 million tonnes of food waste or at least 2.1 million tonnes of avoidable food waste will have to be reduced at the retail (including other distribution of food) and consumer levels by 2030.

The household food waste quantities presented in Table 2 differ significantly from the results of a representative diary-based study (cf. Figure 4 and [15]). The diary-based study estimated household food waste in Germany, excluding sewers, as approximately 3.8 million tonnes (≈47 kg/(cap·year)), a difference of about 2.4 million tonnes compared with our estimates (approx. 6.2 million tonnes or ≈75 kg/(cap·year)). The comparison demonstrates that extrapolations from diary surveys (Scenario II, Figure 4) are more than one-third lower than those from the combined method (Scenario I, Figure 4).

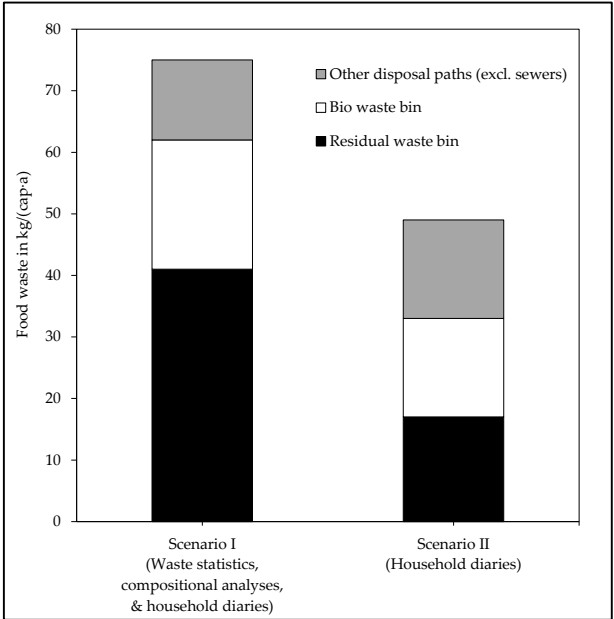

**Figure 4.** Food waste quantities in German households. The mean values of two scenarios are plotted in kg/(cap·year). The estimates of Scenario I are based on a combined method using waste statistics, compositional analyses, and household diaries. The estimates of Scenario II [15] are based on household diaries.

### 3.3. Summary: Food Waste Quantities in Germany

The food waste quantities in Germany and their avoidable parts are presented in Table 3 for each sector of the food supply chain, namely primary production, processing and manufacturing, retail and other distribution of food, restaurants and food services, and households. Accordingly, the total amount of food waste in 2015 was approx. 11.9 (±2.4) million tonnes (excluding sewers in households), of which approx. 6.6 (±1.4) million tonnes can theoretically be avoided. About 12% of food waste occurred during post-harvest processes in primary production and another 18% during processing and manufacturing of food. At the German consumer level, comprising restaurants and food services as well as households, an average of around 66% by mass of food waste was generated (restaurants and food services, 14%; households, 52%). Retail and other distribution of food (including wholesale markets) accounted for the smallest share of food waste, at around 4% by mass.

**Table 3.** Food waste in Germany 2015 in metric tonnes of fresh mass. Summary of results for different sectors of the food supply chain according to Delegated Decision 2019/1597.

|  | **Food Waste** | | | **Theoretically Avoidable Parts** | | |
|---|---|---|---|---|---|---|
|  | Mio. t | kg/(cap·year) | Share | Mio. t | kg/(cap·year) | Share |
| Primary production | 1.4 ± 0.5 | 16.6 ± 6.0 | ≈12% | 1.2 ± 0.4 | 14.2 ± 4.4 | ≈18% |
| Processing and manufacturing | 2.2 ± 0.8 | 26.4 ± 9.1 | ≈18% | 1.2 ± 0.4 | 14.5 ± 5.0 | ≈18% |
| Retail and other distribution of food * | 0.5 ± 0.2 | 6.1 ± 1.9 | ≈4% | 0.4 ± 0.1 | 5.1 ± 1.6 | ≈6% |
| Restaurants and food services | 1.6 ± 0.3 | 19.9 ± 3.4 | ≈14% | 1.1 ± 0.2 | 13.1 ± 2.9 | ≈17% |
| Households (excluding sewers) | 6.2 ± 0.6 | 75.2 ± 7.7 | ≈52% | 2.7 ± 0.3 | 33.1 ± 3.9 | ≈41% |
| ∑ Food waste in Germany | 11.9 ± 2.4 | 144.2 ± 28.1 | 100% | 6.6 ± 1.4 | 80.0 ± 17.8 | 100% |

* including wholesale markets.

Figure 5 provides a graphical overview of the food waste quantities and its theoretically avoidable parts in Germany for the reference year 2015.

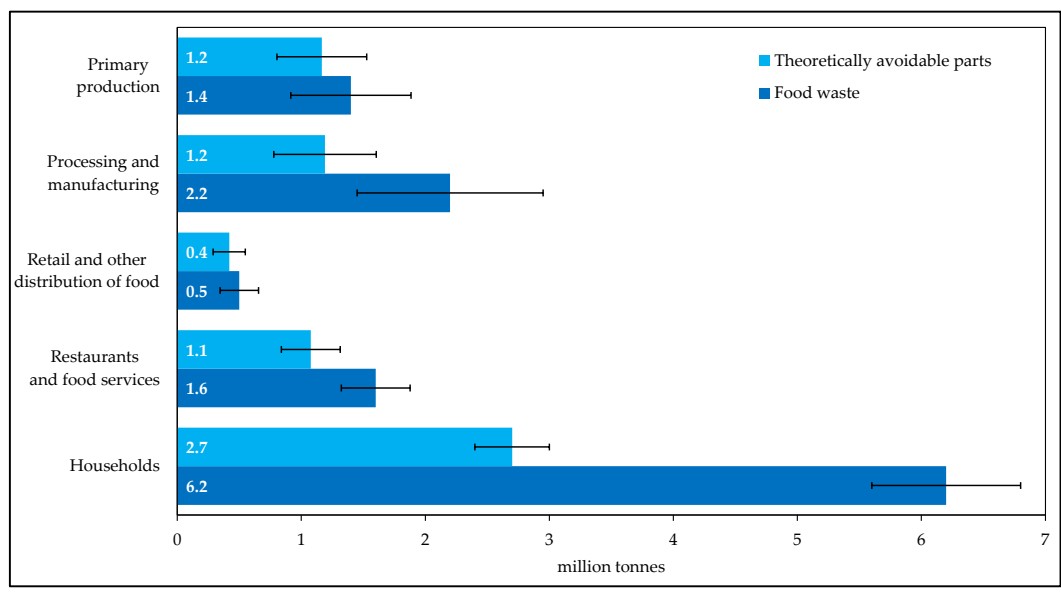

**Figure 5.** Food waste quantities and its avoidable parts in Germany for the reference year 2015 in million metric tonnes of fresh mass (excluding sewers).

## 4. Discussion

### 4.1. General Considerations and Limitations of the Study

The study's outcomes provide a basis for the reporting and monitoring of food waste in Germany within the scope of European legal requirements. With regard to Delegated

Decision (EU) 2019/1597, which requires in-depth measurements in four-year intervals for each sector of the value chain, reliable data are still lacking for determining food waste quantities in Germany on a statistically representative basis. Although our study is based on the best available data at the time, it needs to be acknowledged that several results rely on insufficient data with non-representative sample sizes. An exception was the data sources at the consumer level, for example, with a representative sample size for private households (approx. 7000 households) and reliable statistical data on biodegradable waste from restaurants and food services [15,47]. As such, the results must be interpreted with caution, as they are subject to different degrees of statistical uncertainties throughout the sectors of the food supply chain. Compared with the study of Kranert et al. the food waste estimates ranged within a similar level, but none of the investigated sectors showed a decreasing food waste trend. Slight changes in the estimates since 2012 can be explained by the underlying assumptions and non-representative samples. However, there were no comparable data from 2012 regarding the food waste quantities in primary production. However, the overall food waste quantity across all sectors of the German food supply chain was approx. 11.9 million tonnes (excluding sewers) with an approximate confidence interval of ±2.4 million tonnes (or ±20%). Thus, our results are within a similar range as those of Caldeira et al., who applied two different approaches using material flow analyses (12.37 million tonnes) and the evaluation of waste statistics (11.48 million tonnes) [48]. Furthermore, our dataset was also used by UNEP to prepare the UNEP Food Waste Index Report and to classify the quality of available data for each country. As one of 17 countries worldwide, our German dataset was rated as high confidence by the UNEP, and with high data quality compatible with SDG 12.3.1(b) reporting [29].

### 4.2. Data Quality and Data Gaps

### 4.2.1. Production and Supply Chains

Due to insufficient data sources in harvesting and waste statistics, coefficients from case studies or non-representative surveys had to be used to estimate the food waste quantities in Germany. However, most of the data sources were not collected in the 2015 reference period, further increasing the uncertainty of the results and therefore preventing a representative analysis. Consequently, the current data in primary production do not permit a representative analysis.

Similar problems regarding data sources are evident for the processing and manufacturing sector. The results do not meet the requirements of a statically representative outcome. Uncertainties are due to official waste balance sheets, which do not enable the derivation of any reliable conclusion regarding food waste quantities. For instance, the databases of EUROSTAT and the Federal Statistical Office (DESTATIS) annually provide organic waste quantities from food processing and manufacturing processes for each sub-sector under a specific waste code [49]. Although this issue affects the whole official waste statistics, it is specifically relevant for processing and manufacturing because a variety of branches and implemented technologies affect food waste coefficients. Due to the lack of information on the share of food waste in total (organic) waste streams, the EUROSTAT food waste plug-in, which has been used for preliminary estimation of total food waste in EU member states since 2014, will be phased out in 2022 [5]. It would be an important knowledge gain if official waste statistics were to distinguish between food waste and other organic waste. Therefore, we are convinced that official statistics can be improved with a reasonable effort to provide information on food waste by building on existing sector-specific waste codes. However, the potential of the practical implementation of more specific official waste codes addressing food waste is considered low.

Food waste disposed down the drain, redistribution of surpluses for human consumption, as well as by-products used for animal feeding purposes are excluded from obligatory food waste reporting according to Delegated Decision (EU) 2019/1597 but may be included on voluntary basis. Although we tried our best to provide information about this issue separately, we failed due to data gaps.

In the future, however, a more accurate data basis is expected through voluntary commitments such as the agreement on the reduction in food waste between the German Federal Ministry of Food and Agriculture (BMEL) and associations representing German agriculture (primary production), and the food and nutrition industry [50].

4.2.2. Retail and Consumer Levels

Uncertainties in the results relate to insufficient data sources and a lack of information on food waste from retailers in official waste statistics. For instance, the food waste estimates in retail are largely based on a non-representative evaluation of data from 77 stores of a full-range food retailer and could therefore only provide a non-representative approximation of the true quantities. In comparison with the other sectors, food retail and wholesale markets demonstrated the lowest overall food waste quantities in Germany (499,000 ± 155,000 tonnes in 2015) in relation to other levels of the food supply chain. However, system boundaries in the retail sector were challenging to define. For example, food batches that retailers returned to the manufacturer due to quality flaws were not included in the retail food waste quantities, but were allocated to food waste quantities in the production and supply chain. Furthermore, the present food waste estimates for the German food wholesale only cover the fresh market segment. Data sources for other segments of the German wholesale market (e.g., wholesale related to beverages, processed and convenience food products, or logistical wholesale) are still missing. Thus, the generation of a more complete database for the retail and wholesale sector is part of ongoing research and will be supported by voluntary stakeholder commitments in Germany [51]. With reference to the UN definition, wholesale should be included in the accounting of food losses, whereas in Europe, it is included in retail and other distribution of food.

Since donated food is not considered food waste by definition (cf. Delegated Decision (EU) 2019/1597), we report food donations given by retailers to charities separately from the food waste quantities. However, the retail sector is an important protagonist in the food supply chain that strongly influences the upstream sectors (primary production, processing, and manufacturing), e.g., through the formulation of quality standards. Moreover, retail shapes the purchasing behavior of consumers and their handling of food, e.g., through selling strategies, bundle sizes of food products, packaging, and consumer information. Hence, reduction strategies should consider holistic approaches, involving stakeholders from different sectors, to benefit from the impact of retailers on the food waste generation across food systems.

Biodegradable waste from restaurants and food services was reported in the waste balance sheets of the Federal Statistical Office under the waste code 200,108 [40]. Hence, the statistical data basis is more robust compared with those of other sectors such as primary production, processing and manufacturing, or retail. In addition, data quality shows an improving trend, since several studies and initiatives such as United Against Waste Germany have examined food waste quantities and the reduction potentials for restaurants and food services [16]. To estimate the quantities of food waste that are not reported in official waste balance sheets (e.g., beverages and liquid food waste such as soups and sauces that were disposed through the sewer system), we used findings from the literature. The combination of waste statistics and findings from the literature enabled us to present a more accurate approximation of the true food waste quantities in restaurants and food services and providing data to be reported obligatorily as well as voluntarily.

In households, the food waste quantities were calculated using a method combining official waste statistics, information from diary-based studies, and other household surveys. The information from household diaries represented the food waste percentage that is not disposed of via the municipal waste collection system but ends up in other disposal paths instead. Richter and Bokelmann stated that diaries are suitable for obtaining detailed information about food handling and food waste behavior. However, an uninfluenced survey is impossible when involving the population, and that this can lead to significant errors and misinterpretations in quantification procedures [52]. Participants in diary-based studies

showed a tendency to document significantly less food waste than actually occurred [53,54]. In our study, this effect is also clearly visible when comparing two scenarios (cf. Figure 5). Scenario I contained a broader data basis considering all data sources available. Even though Scenario II was based on a representative sample size (n ≈ 7000), the diary study seemed to significantly underestimate the food waste quantities by approx. 2.4 million tonnes. These findings are in line with those reported by Quested et al., who showed that combining information from waste compositional analyses and household diaries provides more robust estimates, leading to a more accurate approximation of the true food waste quantities in households [55]. A more detailed discussion on this issue using the raw data set of Hübsch and Adlwart was provided by Herzberg et al. [56]. Furthermore, a nationwide and representative compositional analysis of municipal household waste in Germany recently confirmed the findings of Scenario I [57]. According to Dornbusch et al., kitchen food waste amounts to 35.2 kg/(cap·year), while wasted packed food equals 9.3 kg/(cap·year). In total, wasted food in German household residual waste is 44.5 kg/(cap·year) in comparison with the approx. 41 kg/(cap·year) obtained using Scenario I in the present study. Thus, the approach to using municipal waste statistics as basis for food waste quantification in combination with diaries for relevance of other disposal paths is underlined. Given the extensive data sources, we were thus able to present a robust estimation of food waste quantities in German households.

*4.3. Addressing the Research Questions*

4.3.1. Food Waste Reduction Potentials in Germany in the Scope of SDG 12.3

The overall food waste reduction potential in Germany is approx. 11.9 (±2.4) million tonnes, of which 6.6 (±1.4) million tonnes is theoretically avoidable or still suitable for consumption. Households accounted for the highest share of avoidable food waste. In contrast with the U.K., for example, household food waste in Germany has remained more or less at the same level since 2012 [7]. Notably, although British households have reduced their amount by approx. 24% since 2007, consumers in the U.K. (≈100 kg/(cap·year)) still discard significantly more food waste than those in Germany (≈75.2 kg/(cap·year)). Reducing food waste in German households may therefore require different approaches, as the quantities are already equal to the average global food waste level of 74 kg/(cap·year) calculated by the UNEP in 2021 [29]. Considering these findings, the question arises as to whether awareness-raising campaigns and initiatives in Germany would have a similar positive effect as in the U.K. Therefore, the potential impact of consumer-related measures in Germany requires further investigation, especially the feasibility of achieving the political targets. However, compared with the study by Kranert et al., our results showed no statistically relevant reduction in food waste along the different sectors of the food value chain. To achieve SDG 12.3, it is therefore not only necessary to generate reliable databases, but also to develop prevention measures and implement them in the near future.

4.3.2. Monitoring and Reporting of Avoidable Food Waste

According to the Delegated Decision, EU member states may report "amounts of food waste regarded as composed of parts of food intended to be ingested by humans" on a voluntary basis. As the definition of those amounts depends on many individual perceptions, a harmonized definition on EU level has failed so far. Within the present study, we calculated the avoidable amounts of food waste on a separate basis. Although all data originated from Germany, there is no harmonized definition available or applicable. For primary production we used a data base that lists mass percent averages of peels, bones, and kernels [58]. However, this does not answer the question as to whether butternut squash peels, for example, are perceived to be edible in practice and thus should be classified as avoidable or not. For processing and manufacturing, the reasons for food waste mentioned in the conducted survey were the basis for distinguishing the avoidable share. Although the assumption was constructed to the best of our knowledge, there was insufficient information on the amounts of peels, bones, and kernels included in avoidable

food waste during processing and manufacturing. In parallel, estimates for food waste levels in retail, restaurants, and food services are based on non-representative findings and should be improved in the future.

In December 2016, the Saxony State Office for the Environment, Agriculture, and Geology published guidelines with a common methodology to determine food waste percentages in municipal waste collection systems, amending the waste compositional guidelines at the national level [59]. This amendment to the waste compositional guideline is a non-binding recommendation for conducting waste compositional analyses and provides an approach for a standardized method in Germany. In accordance with the amended guideline, food waste is divided into avoidable and unavoidable components. For instance, food that was once edible prior to disposal is considered avoidable food waste. Hence, avoidable food waste includes products such as packaged food with an expired best before date or leftovers. Inedible parts of food products such as bones and peels are considered unavoidable food waste. Based on the mentioned waste compositional guideline, future waste compositional studies may be conducted with a harmonized method and thus contribute to improving the data basis for food waste and avoidable share within German municipal waste collection systems. For this, the broadest possible application in practice at the municipal and federal state level is desirable. Against this background, an even higher degree of information can be expected with regard to future waste classification analyses concerning food waste percentages.

However, the food waste estimates in our study reveal the possibility to indicate the reduction potential of the total as well as edible and avoidable parts of food waste quantities across the entire food supply chain. Based on these findings, we recommend monitoring both the total quantity of food waste and its avoidable components separately by applying a comparable definition of avoidability across the supply chain as much as possible. This recommendation can also be found for Finland, where a clear focus is set on monitoring and preventing edible food waste [60]. Additionally, avoidable food waste quantities may be used to formulate reduction targets and promote a consensus between the political target of halving food waste and the feasibility of reducing food waste both in supply chains and at the consumer levels. A higher acceptance of the formulated reduction targets among stakeholders and households may result in stronger willingness and motivation to reduce food waste.

### 4.3.3. Different Disposal Paths and Alternative Uses

With regard to food waste generated in households, the Delegated Decision excludes food waste that is disposed of via sewers (paragraph 10 of the Delegated Decision (EU) 2019/1597) from the reporting requirements. However, member states have the choice to report information about food waste that ends up in sewers on a voluntary basis. Accordingly, we presented a holistic picture of household food waste quantities in our study by including the disposal paths of residual and biowaste bins, home composting, sewers, and pet feed. Given the still-inconsistent data situation, we considered the EU Commission's approach to give member states the option of documenting food waste streams in other disposal paths on a voluntary basis to be a practical and adequate solution until sufficient methodological experience has been gained to also include these quantities in the reporting. This procedure offers the possibility of obtaining the most complete picture of food waste streams in terms of quality and availability of data.

### 4.3.4. Time Series Trends and Monitoring

Our results refer to the reference year 2015 and represent a basis and starting point for regular reporting at the national level for consecutive calendar years, as required by the European Commission from EU-Member States. In terms of the main goal of the SDG 12.3 (reduction of food loss and food waste in metric tonnes of fresh mass) not the absolute level is crucial, but the proportional advances. This means that time series will be relevant

and trends must be derived. Therefore, it is important to monitor and report trends and progress toward the intermediate EU targets for 2025 and 2030.

According to the "target-measure-act" approach, reporting of food waste generation is only halfway to success. The interpretation of trends is influenced by several variables such as implemented reduction measures, initiatives or other relevant activities, and technical developments. The identification of influencing factors on the overall situation and knowledge-based (re)action makes the monitoring process complete. In addition to the present model, we suggest also considering additional material streams that may support the interpretation of food waste trends in the course of time, such as agricultural commodities, harvesting and yield statistics, by-products, surplus food used for animal feed, and food waste disposed through sewers.

To establish comprehensive monitoring, all relevant food and food waste streams have to be reported. For instance, there is no evidence if a lower food waste amount on the national level was achieved due to real prevention on-site (improved product-waste ratio of a production process) or due to a shift in another use path (innovative use). A comprehensive approach is the key for recommendations for adjustments or further strengthening a promising activity. The Australian national food waste baseline, for example, reports the amounts of food redirected to animal feed or food rescued for human consumption separately for each level of the food supply chain, although those streams are not considered as food waste [61]. Turkey's national food waste prevention strategy also aims to monitor food donations and other forms of redistribution of food for human consumption to implement complete monitoring [62].

The effort for such a monitoring system is derived from in-depth analysis and available data through new surveys and raw data collection, respectively. The system, presented in this paper, has a long history that begun with a study from Kranert et al. [12], continued in the REFOWAS-project [63], and summarized in Baseline 2015 by Schmidt et al. [7]. Monitoring activities are achieved in ongoing dialog forums within the German National Strategy for Food Waste Reduction.

### 4.4. Outlook and Further Steps: National Strategy to Reduce Food Waste in Germany

Our study revealed that data gaps still exist for the primary production, processing and manufacturing, and retail and wholesale sectors. Therefore, cooperation with stakeholders will be necessary to improve data collection with the aim of achieving continuous reporting and monitoring. In this regard, a national food waste reduction strategy was introduced in 2019 by the German Federal Ministry of Food and Agriculture [51]. The strategy represents a central element for meeting SDG 12.3 due to interdisciplinary cooperation in four different fields of action (political framework (1), process optimization (2), behavioral changes among all actors (3), and research (4)). The performance of the strategy will be reviewed five years after its adoption by the federal government. However, the key elements of this strategy are outlined below to provide brief insight into the upcoming measures in Germany and the expected development regarding stakeholder contribution, data quality, and the reduction in food waste.

The political framework was set by the committee of the federal government in cooperation with the federal states' official authorities at different levels with representatives from industry, science, and society. An inter-ministerial Indicator 12.3 Working Group will coordinate food waste reporting in Germany until 2030. In September 2019, a national food waste platform was initiated to establish a stakeholder network for all actors from the food supply chain and civil society. An extension was realized within five sector-specific platforms, in which measures to reduce food waste are developed and voluntary commitments are encouraged [50]. Furthermore, industrial processes shall be improved by involving all stakeholders in the food supply chain who produce, process, handle, or offer food. By reducing food waste, costs will decrease and the sustainable use of resources will be achieved. Proposed measures that can contribute to providing solutions include, for example, analyzing production processes, food waste monitoring, promotion of innovations that

improve processes, cross-sectoral measures, enhancing transparency in the food supply chain, and/or cooperation between interest groups. However, behavioral changes among actors and consumers throughout the food supply chain are addressed by rewarding best practices of effective measures to reduce food waste. Awareness raising activities such as increased communication via social media and training of staff will be implemented. In addition, research and development activities will be funded such as the development of innovative digital solutions for complex logistics, improving the distribution of food donations and smart packaging, and enhancing IT- or AI-supported measuring and forecasting devices.

## 5. Conclusions

Our study represents a starting point for the reporting and monitoring of food waste in Germany. The results can be seen as one option to meet the EU reporting standards based on available data sets supplemented by additional surveys. Through continuous or semi-continuous monitoring of the waste quantities that have been identified, it will be possible to derive trends at the national level. The food waste quantities from the present study should be gradually improved in the future to close data gaps and enable more detailed estimates. The development of indicators adapted to German boundary conditions is part of ongoing work in this area. A representative mapping of the sectors should be the aim, but cannot be ensured in all areas due to the high effort involved on the basis of voluntary agreements. In 2019, a national food waste prevention strategy was launched by the German Ministry of Food and Agriculture, which represents an important step toward implementing a food waste monitoring and achieving reduction targets. In view of the ongoing activities, it can be assumed that the data quality will be improved, for example, through voluntary commitments by important stakeholders along the food supply chain.

**Supplementary Materials:** The following are available online at https://www.mdpi.com/article/10.3390/su13126616/s1. Table S1: Data source of primary production: harvested food quantities ($h_{PP,i}$), food waste coefficients ($wc_{PP,i}$), and share of bones and peels. The food waste coefficients describe the proportion of waste quantities in relation to the quantity of harvested food quantities, Table S2: Data sources for the processing and manufacturing sector: produced food quantities ($p_{PM,i}$) and food waste coefficients ($wc_{PM,i}$). The waste coefficients are based on survey data ($n = 100$) and describe the proportion of waste quantities in relation to the quantity of processed and manufactured food, Table S3: Food waste in German retail: survey data ($n = 77$), Table S4: Data sources for the retail sector: food waste coefficients ($wc_R$). The waste coefficients describe the food waste in kg per square meter of sales area and year, Table S5: Data sources for restaurants and food services: structural data ($s_{RFS,i}$) for different facilities, Table S6: Data sources for restaurants and food services: food waste coefficients ($wc_{RFS,i}$) and avoidable parts, Table S7: Data sources for households: share of food waste ($wc_{HH,i}$) in the municipal waste collection system of residual and biowaste bins in German households, based on data from waste compositional analyses, Table S8: Data sources for households: distribution of food waste through different disposal paths, Figure S1: Drivers that cause food waste in the German food processing and manufacturing sector, based on survey data ($n = 100$). Avoidable potential ($\approx 55\%$): wrong process management (24%), damage and spoilage (14%), overproduction (10%), returns from retailers (6%), and faulty batches (1%).

**Author Contributions:** Conceptualization, D.L., F.S., T.S., G.H. and M.K.; methodology, D.L., F.S., T.S., G.H. and M.K.; software, D.L.; validation, D.L., F.S., G.H., M.K. and Z.N.; formal analysis, D.L.; investigation, D.L.; resources, M.K.; data curation, D.L.; writing—original draft preparation, D.L.; writing—review and editing, D.L. and Z.N.; visualization, D.L.; supervision, M.K. and T.S.; project administration, G.H.; funding acquisition, D.L., G.H. and T.S.; All authors have read and agreed to the published version of the manuscript.

**Funding:** This research was funded by the German Federal Ministry of Education and Research (grant number 01UT1420B) and the German Federal Ministry of Food and Agriculture.

**Data Availability Statement:** The data presented in this study are available in the article and the supplementary materials.

**Acknowledgments:** The authors would like to thank the stakeholders, businesses, companies, food processors and manufacturers, and food retailers for participating in our study by directly providing data or responding to questionnaires and surveys.

**Conflicts of Interest:** The authors declare no conflict of interest. The funders had no role in the design of the study; in the collection, analyses, or interpretation of data; in the writing of the manuscript, or in the decision to publish the results.

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
