# Peer review of "Food Waste Generation in Germany in the Scope of European Legal Requirements for Monitoring and Reporting"

_sustainability, doi:10.3390/su13126616_

Round 1

Reviewer 1 Report

This is a well-written paper with deep analysis. Authors did great effort for collecting and accurately analysing data. There is a weak point of the analysis, that data from many sources were voluntary which is of course a significant distorting effect. However, authors indicate this and do not want to hide it at all. My only recommendation is to change balance between chapters as discussion chapter is too long and I suppose big part of it could be replaced to introduction chapter.

Author Response

Dear Reviewer,

Thank you for the review of our manuscript. On behalf of the co-authors and myself, we hereby send you the revised version of our manuscript.

Referring to the comments, we have made some changes in accordance to the comments of the three reviewers. A summary of the changes is listed below.

All changes are presented as track-changes in the revised document. In addition to the track-change document, we provided a “clean” word document with all changes accepted to better present the new version of the paper.

Thank you in advance for your time and efforts. We look forward to your correspondence regarding the revised manuscript.

Sincerely yours,

Dominik Leverenz

Summary of the changes that we made in accordance with the reviewer comments:

  • We moved “former” chapter 1.4 to the beginning to the introduction (new chapter 1.2).
  • We moved parts from the discussion section (e.g. chapter 4.2.1) into the introduction (compare new chapter 1.2) to shorten the discussion section and get a better balance between the length of the chapters.
  • We have removed chapters 4.2.4 and 4.2.5 to further shorten the discussion. Since the deleted discussion chapters are not directly related to our results, we believe that the paper does not lose content by their removal. We removed two research questions accordingly.
  • In the objective (chapter 1.6) we tried to be more precise in order to highlight the study’s main objective: “With the present calculation of food waste quantities in Germany, we aim to provide a starting point for the future reporting and monitoring of food waste towards 2030. The main objective of the study is to present food waste quantities for each sector of the supply chain according to Delegated Decision (EU) 2019/1597, which requires member states to conduct in-depth measurements every four years. In order to assess the quality of the best available data for the reporting and monitoring process in Germany, we report statistical uncertainties and identify data gaps that need to be filled in the future.
  • In the discussion chapter 4.1., we now clearly present the limitations of the study.
  • In discussion chapter 4.2., we address one of the main objectives of the study by discussing the quality, data gaps and statistical uncertainties.
  • Discussion chapter 4.3 addresses now each of the 4 research questions.

Reviewer 2 Report

My congratulations on this work is well constructed. The methodology is clear, the results are well explained. The conclusions are present, summarize the key findings, underscore your new significant insights, and, then, build on these to develop a solid set of conclusions that come out of your unique study. However, I do have one small comment Authors don’t present limitations to this research. Please complete this.

Author Response

(The authors gave the same response as above.)

Reviewer 3 Report

Authors assessed food waste in the food supply chain in Germany. It was interesting to read this study. The paper is well constructed and clearly written. However, I am not at all convinced with the methodologies used to estimate FW. They are approximations based on non-representative studies. We cannot extrapolate the results of non-representative studies. The results of this paper may be consistent with the reality but they may also be very far from the reality. I understand that the authors opted for these methodologies because data is messing, but I think we cannot draw conclusions from non- rigorous methodologies.

Author Response

Dear Reviewer,

Thank you for the review of our manuscript. On behalf of the co-authors and myself, we hereby send you the revised version of our manuscript.

Referring to the comments, we have made some changes in accordance to the comments of the three reviewers. A summary of the changes is listed below.

All changes are presented as track-changes in the revised document. In addition to the track-change document, we provided a “clean” word document with all changes accepted to better present the new version of the paper.

Thank you in advance for your time and efforts. We look forward to your correspondence regarding the revised manuscript.

Sincerely yours,

Dominik Leverenz

Summary of the changes that we made in accordance with the reviewer comments:

  • We moved “former” chapter 1.4 to the beginning to the introduction (new chapter 1.2).
  • We moved parts from the discussion section (e.g. chapter 4.2.1) into the introduction (compare new chapter 1.2) to shorten the discussion section and get a better balance between the length of the chapters.
  • We have removed chapters 4.2.4 and 4.2.5 to further shorten the discussion. Since the deleted discussion chapters are not directly related to our results, we believe that the paper does not lose content by their removal. We removed two research questions accordingly.
  • In the objective (chapter 1.6) we tried to be more precise in order to highlight the study’s main objective: “With the present calculation of food waste quantities in Germany, we aim to provide a starting point for the future reporting and monitoring of food waste towards 2030. The main objective of the study is to present food waste quantities for each sector of the supply chain according to Delegated Decision (EU) 2019/1597, which requires member states to conduct in-depth measurements every four years. In order to assess the quality of the best available data for the reporting and monitoring process in Germany, we report statistical uncertainties and identify data gaps that need to be filled in the future.
  • In the discussion chapter 4.1., we now clearly present the limitations of the study.
  • In discussion chapter 4.2., we address one of the main objectives of the study by discussing the quality, data gaps and statistical uncertainties.
  • Discussion chapter 4.3 addresses now each of the 4 research questions.

Detailed response to comments of reviewer 3:

01 Reviewer comment regarding methods:

“Authors assessed food waste in the food supply chain in Germany. It was interesting to read this study. The paper is well constructed and clearly written. However, I am not at all convinced with the methodologies used to estimate FW.”

01 Response:

Thank you for this advice. The used methods are defined in European legislation – Delegated Decision (EU) 2019/1597 (Waste framework directive). The EU obliges member states to calculate food waste quantities based on a mix of methods. Our study uses the European legislation as a binding framework. Consequently, all used methods are in line with official requirements. We also criticize the framework, so it is now important to highlight benefits and weaknesses in order to make progress in the monitoring process. Not only Germany is facing these problems, but all European member states. Thus, it is important (and obligatory) to conduct these in-depth measurements and to face out the uncertainties to achieve improvements in the ongoing process.

02 Reviewer comment regarding non-representative studies

“They are approximations based on non-representative studies.”

02 Response

According to Delegated Decision (EU) 2019/1597, samples should meet the requirements statistical representativeness. If this is not possible, Delegated Decision (EU) 2019/1597, demands to use “best available data”. In our study, we used all available data to estimate food waste in Germany and selected the best available data sets in accordance to Delegated Decision (EU) 2019/1597.

03 Reviewer comment regarding extrapolation of non-representative studies

“We cannot extrapolate the results of non-representative studies. The results of this paper may be consistent with the reality but they may also be very far from the reality. I understand that the authors opted for these methodologies because data is messing, but I think we cannot draw conclusions from non- rigorous methodologies.”

03 Response

For each sector we compare our data with the literature and also with studies that used “rigorous” methodologies such as Cladeira et al. (2021).

“However, the overall food waste quantity across all sectors of the German food supply chain resulted to approx. 11.9 million tonnes (excl. sewers) with an approximate confidence interval of ± 2.4 million tonnes (or ± 20%). Thus, our results are within a similar range compared to the study of Caldeira et al. (2021), who applied two different approaches using material flow analyses (12.37 million tonnes) and the evaluation of waste statistics (11.48 million tonnes) [45].”

Furthermore, we verified and critically discussed each data set and result for each sector (see chapter 4.1 and subchapters). Our dataset was also used by UNEP (2021) to prepare the UNEP Food Waste Index Report and to classify the quality of available data for each country. As one of 17 countries worldwide, our German dataset was rated as high confidence by UNEP, with high data quality compatible with SDG 12.3.1(b) reporting.

Despite of that, best available data were used and representative sources were, for example, available for the extrapolations in households and gastronomy.

  • “Biodegradable waste from restaurants and food services was reported in waste balance sheets of the Federal Statistical Office under the waste code 200108 [37].”
  • “To estimate food waste in other disposal paths, information from a diary-based study with a representative panel (≈7,000 German households) was used. Participants had to estimate the type and quantity of food waste and decide whether the waste was avoidable or unavoidable based on their individual and subjective perceptions.”
  • “In order to verify our findings in households, we calculated two scenarios. First, we used a combined method to estimate the food waste quantities in households: Waste statistics and information from waste compositional analyses were used to calculate the quantity in municipal collection while the information from household diaries provided insights on relevance of the additional disposal paths. Thus, food waste in German households was calculated using equation (5) for Scenario I.”
  • “Furthermore, a nationwide and representative compositional analysis of municipal household waste in Germany recently confirmed the findings of Scenario I [54]. According to Dornbusch et al. (2020), kitchen food waste summarizes to 35.2 kg/(cap·a), while wasted packed food equals to 9.3 kg/(cap·a). In total, wasted food in German household residual waste sums up to 44.5 kg/(cap·a) in comparison to approx. 41 kg/(cap·a) comprised by using scenario I in the present study. Thus, the approach to use municipal waste statistics as basis for food waste quantification in combination with diaries for relevance of other disposal paths was underlined. On account of the extensive data sources, we were thus able to present a robust estimation of food waste quantities in German households.”

Round 2

Reviewer 3 Report

Thank you to the authors for the clarifications. 

Author Response

Dear Reviewer,

thank you for your comments.

we have made some further changes in section 1.2 and used a proof reading service.

Attached, we are sending you the revised manuscript with track changes.

Kind regards

Dominik
